# Ultrasonography in the Differentiation of Endometriomas from Hemorrhagic Ovarian Cysts: The Role of Texture Analysis

**DOI:** 10.3390/jpm11070611

**Published:** 2021-06-28

**Authors:** Roxana-Adelina Ștefan, Paul-Andrei Ștefan, Carmen Mihaela Mihu, Csaba Csutak, Carmen Stanca Melincovici, Carmen Bianca Crivii, Andrei Mihai Maluțan, Liviu Hîțu, Andrei Lebovici

**Affiliations:** 1Histology, Morphological Sciences Department, “Iuliu Hațieganu” University of Medicine and Pharmacy, Louis Pasteur Street, Number 4, 400349 Cluj-Napoca, Romania; roxanalupean92@gmail.com (R.-A.Ș.); carmenmihu@umfcluj.ro (C.M.M.); carmen.melincovici@umfcluj.ro (C.S.M.); 2Obstetrics and Gynecology Clinic “Dominic Stanca”, County Emergency Hospital, 21 Decembrie 1989 Boulevard, Number 55, 400094 Cluj-Napoca, Romania; amalutan@umfcluj.ro; 3Anatomy and Embryology, Morphological Sciences Department, “Iuliu Hațieganu” University of Medicine and Pharmacy, Victor Babes, Street, Number 8, 400012 Cluj-Napoca, Romania; bianca.crivii@umfcluj.ro; 4Radiology and Imaging Department, County Emergency Hospital, Cluj-Napoca, Clinicilor Street, Number 5, 400006 Cluj-Napoca, Romania; csutakcsaba@yahoo.com (C.C.); andrei1079@yahoo.com (A.L.); 5Radiology, Surgical Specialties Department, “Iuliu Hațieganu” University of Medicine and Pharmacy, Clinicilor Street, Number 3-5, 400006 Cluj-Napoca, Romania; 6Obstetrics and Gynecology Clinic II, Mother and Child Department, “Iuliu Hațieganu” University of Medicine and Pharmacy, 21 Decembrie 1989 Boulevard, Number 55, 400094 Cluj-Napoca, Romania; 7Doctoral School, Iuliu Hațieganu University of Medicine and Pharmacy, 400012 Cluj-Napoca, Romania; Liviu.Hitu@umfcluj.ro

**Keywords:** computer-aided diagnosis, endometrioma, hemorrhagic cyst, ultrasonography, texture analysis

## Abstract

The ultrasonographic (US) features of endometriomas and hemorrhagic ovarian cysts (HOCs) are often overlapping. With the emergence of new computer-aided diagnosis techniques, this is the first study to investigate whether texture analysis (TA) could improve the discrimination between the two lesions in comparison with classic US evaluation. Fifty-six ovarian cysts (endometriomas, 30; HOCs, 26) were retrospectively included. Four classic US features of endometriomas (low-level internal echoes, perceptible walls, no solid components, and less than five locules) and 275 texture parameters were assessed for every lesion, and the ability to identify endometriomas was evaluated through univariate, multivariate, and receiver operating characteristics analyses. The sensitivity (Se) and specificity (Sp) were calculated with 95% confidence intervals (CIs). The texture model, consisting of seven independent predictors (five variations of difference of variance, image contrast, and the 10th percentile; 100% Se and 100% Sp), was able to outperform the ultrasound model composed of three independent features (low-level internal echoes, perceptible walls, and less than five locules; 74.19% Se and 84.62% Sp) in the diagnosis of endometriomas. The TA showed statistically significant differences between the groups and high diagnostic value, but it remains unclear if the textures reflect the intrinsic histological characteristics of the two lesions.

## 1. Introduction

Transvaginal ultrasonography (TVUS) is the primary diagnostic modality in investigating endometriomas. Usually, this technique can provide enough information for adequate preoperative planning [1]. However, endometriomas share some imaging features with functional hemorrhagic ovarian cysts (HOCs) [2]. Correctly distinguishing the two lesions is vital not only to avoid unnecessary surgery [3] but also because endometriomas are a sign of the presence of other endometriotic lesions at the pelvic and intestinal levels, which can cause a series of complications [4]. As a result, the imaging distinction between the two lesions strongly impacts the course of both medical and surgical treatment [5].

Since it was first described more than 25 years ago [6], the classic “ground glass” appearance of endometriomas has been constantly reevaluated; because of this, researchers have assigned it different levels of diagnostic utility [7,8,9,10] and even integrated it into decision trees, along with other ultrasonographic, clinical, and biological parameters [11]. The ultrasound (US) appearance of endometriomas is highly variable, being influenced by the time-lapse of blood degradation [12]. Moreover, the US features of endometriomas overlap with other lesions such as dermoids, cystic adnexal lesions complicated by bleeding, and cystic ovarian neoplasms [13]. Often, endometriomas and HOCs are indistinguishable, especially in the early stages [2], as they share the characteristic of a cystic mass with bloody content [14]. For these reasons, recent studies [15] concluded that only 60% of endometriomas larger than 2 cm can be appropriately diagnosed with US.

Furthermore, the variety of ultrasound-based descriptors and scoring systems may cause confusion, particularly because their parameters are changeable, sophisticated, and frequently of arbitrary importance [16,17]. Furthermore, the interpretation of medical images is always subjective and observer-dependent [18].

Computer-aided discrimination (CAD) systems have emerged in recent years and attempt to overcome these limitations and increase confidence in the US detection and characterization of ovarian masses [19]. Some of the recently developed CAD techniques focus on the US identification of ovarian malignancies [20,21,22,23,24] and mostly rely on texture analysis (TA) to autonomously detect the presence of a disease based on grayscale variations within TVUS images [25]. TA is a method for extracting and processing parameters that describe pixel intensity and variation patterns, resulting in a quantitative and comprehensive representation of image content [26,27]. The basic concept of texture analysis of ultrasound images is that a diseased process that affects the tissue produces an altered signal, which gives textural features different values than those of the normal structure [28].

This is the first study to provide an ultrasound-based texture characterization of endometriomas and HOCs. We investigated whether texture parameters could be used as an objective diagnostic criterion for distinguishing between the two lesions and whether these parameters were able to outperform the classic US features.

## 2. Materials and Methods

### 2.1. Study Group

This Health Insurance Portability and Accountability Act–compliant, a single-institution, retrospective pilot study, was approved by the institutional review board, and informed consent was waived because of its retrospective nature. From September 2017 to March 2019, a keyword search (using the terms “hemorrhagic + cyst”, alternatives and combinations) in the imaging database of our institution was conducted to identify TVUS images corresponding to ovarian cystic lesions. The keyword search resulted in 235 image reports. Each report was analyzed by one researcher, who excluded all studies that referred to previously documented ovarian malignant or benign tumors (other than endometriomas) (*n* = 57), lesions that were described as having features strongly suggesting malignancy (*n* = 31), and lesions that measured less than 2 cm (*n* = 28). The medical records of the remaining 119 patients were retrieved from the archive of our healthcare institution and investigated for disease-related data. Further, all patients that were transferred to another institution (*n* = 19) and all lesions that were not removed and did not undergo histopathological analysis were also excluded (*n* = 33). The US examinations of the remaining 67 patients were reviewed by one gynecologist who selected only B-mode images that were not affected by artifacts or technique errors. After applying these criteria, US images from 30 endometriomas and 26 hemorrhagic cysts were selected.

### 2.2. Reference Standard

All included lesions underwent pathological analysis after surgical removal. For endometriomas, 12 lesions were removed and analyzed, along with subsequent diseases (uterine fibromatosis, *n* = 9; adenomyosis, *n* = 3). All HOCs underwent pathological analysis because they were included in the surgical specimen analyzed for another condition (atypical endometrial hyperplasia, *n* = 1; adnexal torsion, *n* = 2; uterine leiomyosarcoma, *n* = 2; mucinous cystadenoma, *n* =2; cystadenofibroma, *n* = 3; serous ovarian carcinoma, *n* = 3; serous cystadenomas, *n* = 3; ovarian teratoma, *n* = 4; uterine fibromatosis, *n* = 6).

For pathological analysis, a solution of 10% buffered formalin was used to fixate the surgical samples overnight. Further, using tissue processors, the samples were embedded in paraffin according to the standard protocol of the pathological anatomy laboratory of our institution. The resulting samples were sectioned at 5 µm and stained with hematoxylin and eosin. All resulting samples underwent examination by a pathologist with 9 years of experience in gynecological disease. Following the analysis workflow, a final diagnosis was possible in all the included cases.

### 2.3. Image Acquisition and Interpretation

All the included images were acquired by four gynecologists with at least 8 years of experience in gynecological ultrasound. All examinations were performed on the same machine (Aplio 300, Toshiba Medical Systems, Tokyo, Japan) using a dedicated endovaginal probe (4–10 MHz).

In the first step of image interpretation, each examination was reviewed by one researcher (R.A.Ș.) who was aware of the patients’ pathological findings, clinical outcomes, and final diagnoses. The medical data were cross-referenced with the images to ensure the selection of only the lesions that underwent pathological analysis. Respective lesions were marked, and only one image that was considered representative from each examination was retrieved and anonymized.

In the second step, the typical greyscale US characteristics of endometriomas (“a cyst with internal homogeneous low-level echoes, a perceptible …, no solid component, and a maximum of five locules for multilocular lesions”), as described by Collins et al. [12], were quantified using anonymized images by another researcher (A.M.M.) who was also blinded to the patients’ outcomes. The homogeneous low-level echoes (or ground glass) appearance was considered if this was the dominant pattern in more than 90% of the lesion’s content. The lesions were considered to have a wall if a structure at least 2 mm thick that surrounded at least 50% of the visible portion of the cyst could be observed. Since only gray-scale images were retrieved, any structure that was adjacent to the walls was considered a solid component (including hyperechoic foci, papillary projections, and retracting clots). Furthermore, unilocular lesions were also considered to have a maximum of five locules.

### 2.4. Statistical Analysis

To quantify the information in a quantitative way, for each lesion, each ultrasound parameter was given the value of “1” if present or “0” if absent. A multiple regression (multivariate) analysis was conducted to investigate which ultrasound features could independently predict the presence of endometriomas. The analysis was conducted using the “enter” input model, which involved entering all variables into the model in one single step. A conventional *p*-value of less than 0.05 was used to determine the corresponding independent variables that contributed significantly to the differentiation of endometriomas from HOCs, whereas variables with a *p*-value of more than 0.01 were omitted. In addition, the coefficient of determination (R^2^, the proportion of the variation in the dependent variable explained by the regression model, measuring of the goodness of fit of the model), the R^2^-adjusted coefficient (the coefficient of determination adjusted for the number of independent variables in the regression model), the multiple correlation coefficient (measuring how tightly the data points clustered around the regression plane, calculated by taking the square root of the coefficient of determination), and the variance inflation factor (VIF, an indicator of multicollinearity) were calculated. After the analysis, the predicted values were saved and then used in a receiver operating characteristic (ROC) analysis to determine the prediction model’s ability to identify endometriomas. The ROC analysis was also used to test the ability of each ultrasound feature in the diagnosis of endometriomas. The DeLong et al. technique was used to compute the ROC curves, and the binomial exact confidence intervals for the areas under the curve (AUC) were stated. The optimal cut-off values for predicting patients with malignancies were determined using a common optimization step that maximized the Youden index (J). Specificity (Sp) and sensitivity (Se) were calculated from the same data, without other adjustments, using a 95% confidence interval (CI).

### 2.5. Texture Analysis Protocol

The radiomics approach consisted of five steps: image pre-processing, lesion segmentation, feature extraction, feature selection, and prediction.

#### 2.5.1. Image Pre-Processing and Segmentation

Images were retrieved in Digital Imaging and Communications in Medicine (DICOM) format and were further converted into Joint Photographic Experts Group format (JPG) and imported into a dedicated software (Topaz DeNoise AI, Topaz Labs, TX, USA) in which the negative impact of the speckle noise was reduced using a denoising technique based on convolutional neural networks [29]. Afterward, images were reconverted into bitmap format and transferred to a dedicated texture analysis software (MaZda, Institute of Electronics, Technical University of Lodz, Lodz, Poland) [30]. Using this program, the image grey levels were normalized based on the mean and three standard deviations of grey level intensities to reduce the contrast and brightness variations.

The image segmentation process was performed by a second researcher (P.A.Ș.) who was blinded to the outcomes of the patients. The researcher incorporated each lesion into a two-dimensional region of interest (ROI). The first step of the ROI definition process was performed semi-automatically. The researcher drew a circle inside each lesion and the software automatically delineated the structure of interest based on gradient and geometry coordinates. In the second step, if a complete overlap between the ROI and the structure’s contours was detected, the ROI was manually adjusted (Figure 1).

#### 2.5.2. Feature Extraction

The texture features (or parameters) were automatically extracted by the software after the definition and positioning of each ROI. From each lesion, a total of 275 parameters were computed [31]. The parameters are described in Table 1.

For each lesion, the segmentation and extraction of texture parameters were repeated 1 week apart, and the process was carried out by the same researcher. The resulting values were used to evaluate the intra-reader agreement using the intraclass coefficient.

#### 2.5.3. Feature Selection

In order to identify the best-suited texture parameters for differentiating between the two histopathological groups, three methods were applied successively. The first step comprised of applying three reduction methods (based on mutual information (MI), Fisher coefficients (F, the ratio of between-class to within-class variance), and the probability of classification error and average correlation coefficients (POE + ACC)) [32]. Each of the three selection methods provided a set of ten unique parameters.

Second, the intraclass correlation coefficient (ICC) was calculated using the absolute agreement between the same rater for the all-subjects model, and average values along with the 95% confidence interval were reported. Features that demonstrated an ICC of below 0.85 were excluded from further analysis.

Third, the absolute values of the remaining parameters were compared between the two groups using the Mann–Whitney U test. The statistically significant level was set at a *p*-value of below 0.0016 after Bonferroni correction (which implied dividing the classic 0.05 level by 30, considering the 27 unique parameters that resulted after applying the reduction techniques as well as age and the two separate histopathological entities). All texture parameters that showed univariate analysis results below this threshold were excluded from further processing.

#### 2.5.4. Class Prediction

To investigate which of the previously selected texture features were independent predictors of endometriomas, a multiple regression analysis was performed following the same computational method as was used for the ultrasound features. Furthermore, features with a VIF greater than 10^4^ were withdrawn from further testing because a high VIF value indicates multicollinearity. The predicted values were saved and then used in an ROC analysis to determine the prediction model’s ability to identify endometriomas. The ROC analysis was also used to test the diagnostic utility of the features that were independently associated with endometriomas, following the same workflow as described for the ultrasound features. Statistical analysis was performed using a commercially available dedicated software, MedCalc version 14.8.1 (MedCalc Software, Mariakerke, Belgium). The workflow model is summarized in Figure 2.

## 3. Results

Fifty-six patients (average age ± standard deviation: 38.27 ± 14.68 years; age range: 22–54 years) were included according to their final diagnosis. Patients were divided into an endometrioma group (*n* = 30) and an HOC group (*n* = 26).

When analyzing the gray-scale features, three out of four characteristics (internal homogeneous low-level echoes, a perceptible wall, and a maximum of five locules) were independently associated with endometriomas (Table 2). The multivariate analysis showed a significance level of *p* < 0.0001, an R^2^ coefficient of determination of 0.3856, an adjusted R^2^ value of 0.3384, and a multiple correlation coefficient of 0.621. The diagnostic performance of the three independent US features and the prediction model is displayed in Table 3.

For the texture analysis, one variation of the difference of variance parameter (CN5D6DifVarnc) was selected by both the Fisher and POE + ACC methods, while two variations of the same feature (CN4D6DifVarnc and CH5D6DifVarnc) were highlighted by both the Fisher and MI methods. In total, 27 unique parameters resulted after applying the three reduction techniques. The results of the univariate analysis and intra-reader agreement evaluation are displayed in Table 4.

Twenty parameters showed statistically significant results in the univariate analysis and underwent multiple regression analysis. The parameter CN4D6DifVarnc was excluded from the analysis because it had a VIF greater than 10^4^. The multivariate analysis showed a significance level of *p* < 0.001, an R^2^ coefficient of determination of 0.435, an adjusted R^2^ value of 0.427, and a multiple correlation coefficient of 0.634. The multiple regression analysis identified seven parameters as independent predictors of endometriomas (Table 5).

The ROC analysis showed that the prediction model exceeded the individual diagnostic ability of all independent features in terms of both sensitivity and specificity (Table 6, Figure 3). The texture maps that display the distribution of selected texture features in images from each entity are shown in Figure 4.

## 4. Discussion

Our results showed that the majority of the included endometriomas (*n* = 23) expressed low-level internal echoes, while this indicator was encountered in less than 30% of HOCs (*n* = 7). As expected, this feature was the most distinctive sign of endometriomas, as previously described since the first research was conducted in the field (Table 7). However, we were unable to find a study that specifically aimed to address the distinction between endometriomas and HOCs based on grey-level ultrasound features, with most research focusing on distinguishing endometriomas from other ovarian tumors (sometimes including HOCs) [7,11]. In our study, the low-level internal echoes (or ground glass appearance) showed similar sensitivity (74.19%) but lower specificity for the diagnosis of endometriomas compared with the most recent research in the field (73% Se; 94% Sp) [11]. Moreover, when other features were added to the model, the overall sensitivity did not increase.

Hemorrhagic ovarian cysts are caused by bleeding inside functional cysts that are spontaneously resorbed [2]. They also progress slowly through different stages of acute hemorrhage, clot development, and retraction, resulting in a shifting sonographic appearance until they fully resolve in 6 weeks or shrink significantly in size [33]. In the early stages, they appear as solid masses with thin walls. Furthermore, their content may express variable echogenicity with reticular strands. When the clot retracts, its attachment to the wall can mimic a papillary projection, and a fluid layer also develops within the cyst [34].

At the beginning of their formation, the US appearance of endometriomas can be indistinguishable from that of HOCs. In time, as the bleeding becomes chronic, endometriomas build up more hemorrhagic debris [2], which is responsible for their classic US appearance (of a “unilocular cyst with fluid content expressing ground glass echogenicity” or “ground glass”) [35,36]. However, further research [11] demonstrated that less than 50% of endometriomas exhibit these characteristics, with even lower rates in the postmenopausal population. This is most likely because endometriomas express cyclic bleeding, which results in different time stages of blood degradation, thereby generating variable US appearance [37].

In practice, the grayscale imaging of endometriomas and HOCs can be identical due to bleeding features of different ages, making distinction difficult [2]. On one hand, sonographic observation of fibrin strands and/or retracting clots within an adnexal cyst reflects a recent episode of hemorrhage [35]. Due to the cyclic bleeding characteristic of endometriomas, the appearance of fibrin stands can easily mimic the features of an HOC [38]. On the other hand, HOCs can also demonstrate diffuse low-level internal echoes [7], most likely because in some cases, they do not regress and instead accumulate various quantities of intracellular deoxyhemoglobin and methemoglobin [39].

Considering the information above, together with the subjective nature of the interpretation of US images, a clear differentiation between the two lesion types can be performed only through microscopic evaluation [40]. The cell population found within endometriomas consists of degenerate erythrocytes, hemosiderin-laden macrophages, and endometrial and epithelial cells [41,42,43,44]. Functional hemorrhagic cysts’ content is composed of a mixture of blood products along with plasma proteins, mucopolysaccharides, and hyaluronic acid [45], and these lesions also have rich cellularity [43]. It is possible that all the histopathological characteristics are reflected within US images, subtly influencing the pixel intensity and distribution, which may be detected and quantified through texture parameters.

Our results showed that five variations of the difference of variance parameter were independent predictors of endometriomas (CN5D6DifVarnc, CV5D6DifVarnc, CZ3D6DifVarnc, CZ4D6DifVarnc, and CZ5D6DifVarnc). The variance is a measure of contrast severity, which can be used to create descriptors of relative smoothness. [46]. The difference of variance measures the variance of the difference of grey level values (reflecting the randomness within an image [47,48]. In all scenarios, this feature exhibited higher values for HOCs than for endometriomas.

The contrast parameter shows the local variations present in an image, expressing higher values when an image contains a large number of pixels with different grey level values [47]. We obtained higher values for the HOCs than for endometriomas.

The first-order histogram parameters (mean, standard deviation, variance, skewness, kurtosis, and percentiles) reflect the value of the pixel intensity, without considering the spatial relations between the pixels [49]. The percentile number (*n*) is the point in the histogram where *n*% of the pixel values are found to the left [50]. A percentile, in other words, is the highest grey level value at which a given percentage of the pixels in an image are contained [51]. This signifies that 10% of the pixels within images were distributed under higher values for endometriomas than for HOCs.

Although it was expected that endometriomas would show a higher degree of echogenic randomness because of the multitude of contained elements, the parameters indicating these characteristics showed higher values for HOCs. This observation is in accordance with the literature, which indicates that HOCs have more complex and heterogenous content on TVUS (probably because they more often express fine linear strands and retracting clots) [52].

In the current study, the texture model was able to diagnose endometriomas with almost perfect rates: 100% Se (CI, 88.4–100%) and 100% Sp (CI, 75.3–100%). In a similar study, based on magnetic resonance (MRI) images [53], the texture model was able to distinguish endometriomas from HOCs with similar rates, showing a sensitivity of 100% (CI, 85.8–100%) and a specificity of 100% (CI, 71.5–100%). Once again, this model [53] outperformed the classic MRI features of endometriomas (“T2 shading”, 75.86% Se and 35.71% Sp and “T2 dark spots”, 55.17% Se and 64.29% Sp). However, the current model comprised different texture parameters (five variations of difference of variance, image contrast, and 10th percentile) compared to those extracted from MRI images (which included mostly variations of entropy) [53]. The high accuracy rates accomplished by both models could indicate that TA is feasible in distinguishing the two lesions in both types of imaging examinations. However, these excellent results may be influenced by the reduction techniques (especially the Fisher method); although they provide the most distinguishing parameters, these parameters are highly correlated [32], and they could therefore influence the diagnostic value of the prediction model [54]. In order to at least partially counteract this effect, in addition to the Fisher method, two other selection methods were used that did not provide parameters with a high degree of correlation (POE + ACC and Mutual Information). Moreover, parameters that showed statistical significance in the univariate analysis but also demonstrated a VIF >10^4^ were excluded from the final model.

Because the ultrasound features of the two lesions may overlap, sometimes the diagnosis cannot be straightforward. Our TA model may be useful in providing more confidence in the diagnosis of a newly discovered bleeding ovarian lesion. Moreover, if this approach is further validated, it could offer an alternative to more expensive and time-consuming approaches for characterizing adnexal lesions, such as MRI examinations. However, the role of MRI in the characterization of endometriomas will not be diminished since some lesions could remain inapparent to TVUS despite the visualization of the ovaries because of their location or the presence of periovarian adhesions [55].

Our study had several limitations. Due to the retrospective design and the decision to include only pathologically proven lesions, the study may possess selection and verification bias. In the final step of patient selection, all lesions below 2 cm (*n* = 33) were excluded. This threshold was necessary to provide a sufficient area for the software to analyze and extract the pixel pattern. Therefore, the study population was relatively low. The small population was also due to the limited time approved for this research by the ethics committee and the strict exclusion and inclusion criteria. The menstrual phase, CA-125 levels, and menopausal status were not reported since they were inconsistently mentioned in the retrieved medical data. Few of the selected patients had Doppler images stored in our database; therefore, only grey scale images were selected. For this reason, other classic features of endometriomas, such as no internal vascularity and avascular internal nodules [12], could not be quantified. However, the use of color Doppler imaging does not improve the diagnostic accuracy of transvaginal ultrasonography alone in the diagnosis of ovarian endometrioma [56]. On the other hand, it is documented that the presence of intracystic vascularization poses doubts about malignancy [57]. However, this was not the case for selected lesions, as the pathological analysis did not raise suspicions of malignancy in any case. A limitation was that one investigator was aware of the final diagnoses of the lesions. However, since many patients had several adnexal lesions at the time of the US evaluation, this strategy was necessary for selecting only documented lesions. This researcher was not involved in the image segmentation, statistical analysis, or reporting of the results after this point. Considering these limitations and the pilot nature of this research, the presented TA model for discriminating endometriomas from HOCs requires prospective research for both validation and establishment of its clinical utility compared with the classic imaging methods.

## 5. Conclusions

We demonstrated a statistically significant difference between the texture features of endometriomas and hemorrhagic ovarian cysts. Although this approach outperformed the classic ultrasound evaluation, it remains unclear if textures reflect the lesions’ pathological characteristics. Further studies are required to identify the exact substrate that determines textural differentiation.

## Figures and Tables

**Figure 1 jpm-11-00611-f001:**
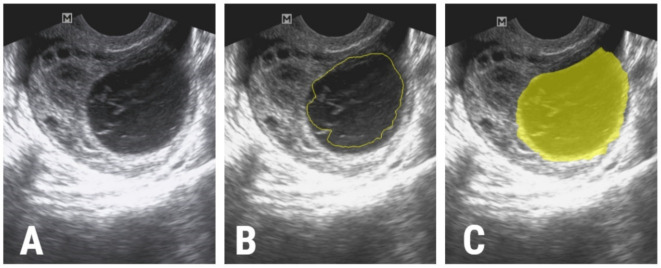
(**A**) The ultrasound image of a 29-year-old patient with s histologically proven hemorrhagic ovarian cyst, (**B**) the initial ROI that was automatically delineated by the software (yellow line), and (**C**) the final ROI after manual adjustments (yellow).

**Figure 2 jpm-11-00611-f002:**
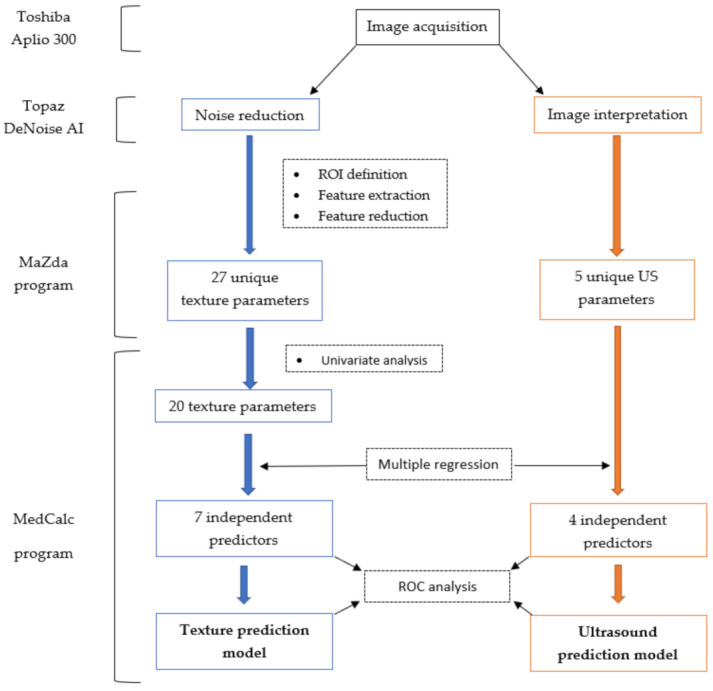
Workflow model summarizing the construction of the texture (blue) and ultrasound (orange) prediction models. US, ultrasound; ROI, region of interest; ROC, receiver operating characteristic.

**Figure 3 jpm-11-00611-f003:**
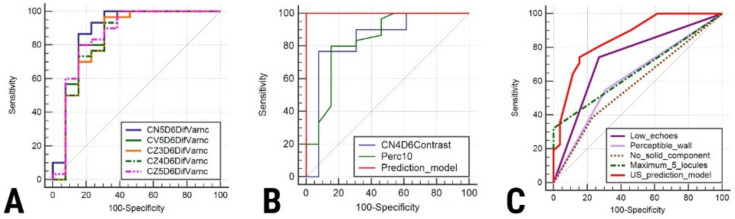
Comparison of areas under the curve for the differentiation of endometriomas from hemorrhagic cysts based on (**A**) the difference of variance parameters, (**B**) contrast and percentile parameters and the prediction model composed of independent texture parameters, and (**C**) classic ultrasound features of endometriomas and their combined diagnostic value.

**Figure 4 jpm-11-00611-f004:**
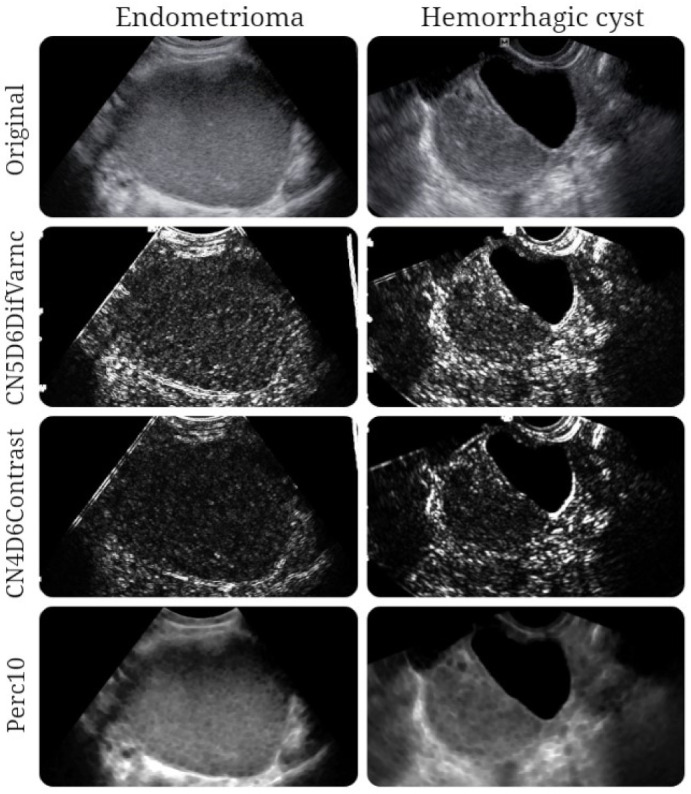
Texture maps showing the distribution of three texture parameters (CN5D6DifVarnc, CN4D6Contrast, and Perc10) in images of patients with a histologically proven endometrioma and hemorrhagic ovarian cyst.

**Table 1 jpm-11-00611-t001:** Texture parameters.

Class	Parameters	Computation	Variations	Number of Parameters
Histogram	Mean, Variance, Skewness, Kurtosis, Perc.01–99%	-	-	5
Absolute gradient	GrMean, GrVariance, GrSkewness, GrKurtosis, GrNonZeros, percentage of pixels with nonzero gradient	4 bits/pixel	-	5
Run Length Matrix	RLNonUni, GLevNonU, LngREmph, ShrtREmp, Fraction	6 bits/pixel	4 directions	20
Co-occurrence Matrix	AngScMom, Contrast, Correlat, SumOfSqs, InvDfMom, SumAverg, SumVarnc, SumEntrp, Entropy, DifVarnc, DifEntrp	6 bits/pixel; 5 between-pixel distances	4 directions	220
Auto-regressive Model	Teta 1–4, Sigma	-	-	5
Wavelet transformation	WavEn	5 scales	4 frequency bands	20

Mean, histogram’s mean; Variance, histogram’s variance; Skewness, histogram’s skewness; Kurtosis, histogram’s kurtosis; Perc.01–99%, 1st to99th percentile; GrMean, absolute gradient mean; GrVariance, absolute gradient variance; GrSkewness, absolute gradient skewness; GrKurtosis, absolute gradient kurtosis; GrNonZeros, percentage of pixels with nonzero gradient); RLNonUni, run-length nonuniformity; GLevNonU, grey level nonuniformity; LngREmph, long-run emphasis; ShrtREmp, short-run emphasis; Fraction, the fraction of image in runs; AngScMom, angular second moment; Contrast, contrast; Correlat, correlation; SumOfSqs, the sum of squares; InvDfMom, inverse difference moment; SumAverg, sum average; SumVarnc, sum variance; SumEntrp, sum entropy; Entropy, entropy; DifVarnc, difference of variance; DifEntrp, difference of entropy; Teta 1–4, parameters θ1–θ4; Sigma, parameter σ; WavEn, wavelet energy.

**Table 2 jpm-11-00611-t002:** The multivariate analysis showing independent features associated with the presence of endometriomas. The cases in which a specific feature could be found in either group are displayed as number/total.

Parameter	Endometriomas	HOCs	Coefficient	Standard Error	*p*-Value	VIF
Internal homogeneous low-level echoes	23/30	7/26	0.3151	0.1177	**0.0099**	1.178
Perceptible wall	17/30	8/26	−0.4162	0.1557	**0.01**	1.197
No solid component	12/30	6/26	0.1977	0.1234	0.1153	1.123
Maximum of 5 locules	21/30	26/26	0.2971	0.1167	**0.0139**	1.144

Bold values are statistically significant (*p* < 0.05). VIF, variance inflation factor.

**Table 3 jpm-11-00611-t003:** The receiver operating characteristic analysis results of the prediction model and the ultrasound parameters that were independently associated with endometriomas.

US Feature	AUC	Sign.lvl.	J	Cut-Off	Se (%)	Sp (%)
Internal homogeneous low-level echoes	0.736 (0.603–0.844)	**0.0001**	0.47	>0	74.19 (55.4–88.1)	73.08 (52.2–88.4)
Perceptible wall	0.62 (0.482–0.746)	0.0631	0.24	>0	54.84 (36–72.7)	69.23 (48.2–85.7)
Maximum of 5 locules	0.661 (0.524–0.781)	**0.0002**	0.32	≤0	32.26 (16.7–51.4)	100 (86.8–100)
US prediction model	0.857 (0.739–0.936)	**<0.0001**	0.58	>0.42	74.19 (55.4–88.1)	84.62 (65.1–95.6)

The values corresponding to 95% confidence intervals are shown in parentheses. Bold values are statistically significant. US, ultrasound; US prediction model, the model composed of the predictive values provided by the multivariate analysis of ultrasound features; Sign.lvl., significance level; J, Youden index; Se, sensitivity; Sp, specificity.

**Table 4 jpm-11-00611-t004:** The univariate analysis (Mann–Whitney U test) and the intra-reader agreement evaluation results.

Parameter	*p*-Value	Endometriomas		Hemorrhagic Cysts		Agreement	
		Median	IQR	Median	IQR	ICC	95% CI
Fisher							
CN5D6DifVarnc	**0.0001**	8.33	4.73–14.81	22.85	18.94–32.58	0.98	0.97–0.99
CN4D6DifVarnc	**0.0002**	7.95	3.95–14.7	20.79	18.27–30.62	0.98	0.98–0.99
CV5D6DifVarnc	**0.0002**	7.77	4.28–15.01	23.61	16.32–28.37	0.98	0.97–0.99
CZ5D6DifVarnc	**0.0002**	8.37	4.75–15.55	23.25	16.6–30.34	0.98	0.97–0.99
CN3D6DifVarnc	**0.0002**	7.04	3.61–14.55	20.82	15.5–28.04	0.93	0.98–0.99
CV4D6DifVarnc	**0.0003**	7.43	3.71–14.88	22.12	15.46–26.6	0.99	0.98–0.99
CZ4D6DifVarnc	**0.0004**	7.81	4.49–15.16	22.67	15.14–27.66	0.99	0.98–0.99
CH5D6DifVarnc	**0.0006**	7.03	3.63–14.39	22.11	13.52–27.51	0.99	0.99–0.99
CZ3D6DifVarnc	**0.0005**	7.38	3.95–14.6	21.67	13.97–25.14	0.99	0.98–0.99
WavEnHL_s-2	**0.0008**	10.65	5.33–20.82	29.68	16.84–38.2	0.99	0.99–0.99
POE + ACC							
RZD6Fraction	0.0445	0.74	0.7–0.8	0.67	0.43–0.76	0.99	0.98–0.99
RVD6GLevNonU	**0.0006**	2849.65	1420.58–3750.88	1081.5	575.26–1718.42	0.99	0.99–0.99
WavEnHL_s-5	0.007	30.72	22.22–61.06	69.11	47.55–124.11	0.99	0.99–0.99
RVD6LngREmph	0.0103	2.27	1.78–2.69	5.67	2.35–37.05	0.98	0.97–0.99
ATeta4	0.5085	0.15	0.01–0.38	0.16	0.12–0.31	0.99	0.99–0.99
GD4Kurtosis	0.2783	10.59	0.1–67.8	48.14	28.16–58.71	0.99	0.98–0.99
Perc10	**0.0005**	31.5	24–39	4	1–19	0.99	0.99–0.99
CN5D6Correlat	0.0569	0.57	0.5–0.71	0.45	0.25–0.59	0.98	0.96–0.98
RZD6GLevNonU	**0.0009**	3041.64	1300.63–3769.19	1079.18	559.01–1829.93	0.99	0.99–0.99
Mutual Information							
WavEnHH_s-3	0.0011	8.46	3.95–10.79	15.6	9.8–19.88	0.99	0.98–0.99
CN3D6Contrast	**0.0005**	15.81	7.45–19.29	28.84	22.47–37.86	0.99	0.98–0.99
WavEnLH_s-3	**0.0008**	23.73	17.75–30.86	46.16	30.64–60.69	0.97	0.94–0.98
WavEnHH_s-4	0.0055	7.18	5.42–12.16	13.69	11.17–17.1003	0.96	0.94–0.98
CH4D6DifVarnc	**0.0009**	6.18	3.2–14.51	20.33	12.47–24.73	0.99	0.99–0.99
CV5D6Contrast	**0.0007**	16.91	9.03–21.58	33.39	23.8–40.08	0.98	0.97–0.99
CN4D6Contrast	**0.0005**	17.49	8.44–21.16	31.49	23.18–39.65	0.98	0.97–0.99
CH4D6Contrast	0.0033	13.22	6.17–17.78	26.23	16.25–33.47	0.99	0.99–0.99

Statistically significant results from the Mann-–Whitney U-test are highlighted in bold. IQR, interquartile range; POE + ACC, probability of classification error and average correlation coefficient; ICC, intraclass coefficient; DifVarnc, difference of variance; WavEn, wavelet energy; Fraction, the fraction of image in runs; GLevNonU, grey level nonuniformity; GLevNonU, long-run emphasis; Teta, parameter θ4; Kurtosis, histogram’s kurtosis; Perc10, 10th percentile; Correlat, correlation; Contrast, contrast.

**Table 5 jpm-11-00611-t005:** Multivariate analysis results showing the texture parameters independently associated with the presence of endometriomas.

Parameter	Coefficient	Standard Error	*p*-Value	VIF
CH4D6DifVarnc	0.2662	0.1931	0.1808	2965.650
CH5D6DifVarnc	−0.2735	0.1944	0.1722	3131.661
CN3D6DifVarnc	−0.04859	0.2082	0.8174	3440.996
CN5D6DifVarnc	0.4225	0.1255	**0.0026**	1425.172
CV4D6DifVarnc	0.01293	0.2247	0.9546	4143.205
CV5D6DifVarnc	−0.5476	0.1931	**0.0091**	3225.142
CZ3D6DifVarnc	−0.4053	0.1928	**0.0462**	3080.372
CZ4D6DifVarnc	1.0179	0.2813	**0.0014**	6825.702
CZ5D6DifVarnc	−0.4797	0.1352	**0.0016**	1621.315
CN3D6Contrast	0.2624	0.1357	0.0651	2136.581
CN4D6Contrast	−0.4877	0.1945	**0.0193**	5009.060
CV5D6Contrast	0.2167	0.1179	0.0784	1795.666
Perc10	0.01755	0.004113	**0.0003**	3.535
RVD6GLevNonU	0.0002	0.0002	0.4101	123.113
RZD6GLevNonU	−0.0001	0.0002	0.6515	125.807
WavEnHH_s_3	0.01052	0.02963	0.7255	29.482
WavEnHL_s_2	0.017	0.01602	0.2993	37.886
WavEnLH_s_3	0.008	0.006074	0.1791	7.886

Bold values are statistically significant (*p* < 0.05). VIF, variance inflation factor.

**Table 6 jpm-11-00611-t006:** The receiver operating characteristic analysis results of the prediction model and the texture parameters that were independently associated with endometriomas.

Parameter	AUC	Sign.lvl.	J	Cut-Off	Se (%)	Sp (%)
CN5D6DifVarnc	0.882 (0.747–0.960)	**<0.0001**	0.7128	≤16.73	86.67 (69.3–96.2)	84.62 (54.6–98.1)
CV5D6DifVarnc	0.856 (0.716–0.944)	**<0.0001**	0.659	≤21.26	96.67 (82.8–99.9)	69.23 (38.6–90.9)
CZ3D6DifVarnc	0.838 (0.694–0.933)	**0.0001**	0.659	≤20.12	96.67 (82.8–99.9)	69.23 (38.6–90.9)
CZ4D6DifVarnc	0.841 (0.697–0.934)	**0.0001**	0.625	≤19.43	93.33 (77.9–99.2)	69.23 (38.6–90.9)
CZ5D6DifVarnc	0.859 (0.719–0.946)	**<0.0001**	0.6462	≤15.67	80 (61.4–92.3)	84.62 (54.6–98.1)
CN4D6Contrast	0.838 (0.694–0.933)	**<0.0001**	0.6897	≤21.16	76.67 (57.7–90.1)	92.31 (64–99.8)
Perc10	0.836 (0.691–0.931)	**<0.0001**	0.6462	>19	80 (61.4–92.3)	84.62 (54.6–98.1)
Texture prediction model	1 (0.918–1)	**<0.0001**	1	>0.4	100 (88.4–100)	100 (75.3–100)

The values corresponding to 95% confidence intervals are shown in parentheses. Bold values are statistically significant. Texture prediction model, the model composed of the predictive values provided by the multivariate analysis of the texture features; Sign.lvl., significance level; J, Youden index; Se, sensitivity; Sp, specificity.

**Table 7 jpm-11-00611-t007:** Research involving endometriomas’ classic ultrasound features.

Imaging Feature	Author/Year	Diagnostic Value
Low-level internal echoes	Patel et al. (1999) [7]	95% Se; 49% Sp
Low-level internal echoes, no neoplastic features, and no fibrinous strands or retracting clots	Patel et al. (1999) [7]	65% Se; 76% Sp
Low-level internal echoes, no neoplastic features, and no hyperechoic wall foci	Patel et al. (1999) [7]	30% Se; 86% Sp
Low-level internal echoes, no neoplastic features, and hyperechoic wall foci or multilocularity	Patel et al. (1999) [7]	45% Se; 90% Sp
“Round, intraovarian, homogeneous, hypoechoic tissue, with a clear demarcation from the parenchyma and without papillary proliferations”	Mais et al. (1993) [6]	84% Se; 90% Sp
“A cystic structure with low, homogeneous echogenicity and a thick cystic wall with regular margins” but not excluding “very fine papillary structures, not exceeding 3 mm”	Volpi et al. (1995) [8]	82.4% Se; 97.7% Sp
“The presence of a round-shaped homogeneoushypoechoic mass of low-level echoes”	Alcazar et al. (1997) [9]	88.9% Se; 91% Sp
“Round-shaped homogeneous hypoechoic ‘tissue’ of low-level echoes within the ovary”	Guerriero et al. (1996) [10]	79% Se; 76% Sp
Ground glass echogenicity of cyst fluid	Van Holsbeke et al. (2010) [11]	73% Se; 94% Sp

Se, sensitivity; Sp, specificity.

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
