# Peer review of "Ultrasonography in the Differentiation of Endometriomas from Hemorrhagic Ovarian Cysts: The Role of Texture Analysis"

_jpm, 2021, doi:10.3390/jpm11070611_

Round 1
Reviewer 1 Report
Dear editor:
Thank you for inviting me to evaluate this article titled “Ultrasonography in the Differentiation of Endometriomas from Hemorrhagic Ovarian Cysts: the Role of Texture Analysis”. The authors proposed models to test the statistically significant difference between the texture features of endometriomas and hemorrhagic ovarian cysts (HOC). This manuscript is of great interest to researchers. Much of the manuscript has been well characterized, but there are some major and minor concerns that need to be addressed before they can be published.
Major comments:
- In Table 5, the VIFs of most parameters are extremely high, suggesting that some of the parameters are at least marginally correlated. The authors can optimize their model by filtering the selected parameters. I believe some of the parameters can be removed without affecting the performance of the model.
- [Line 143]: “To identify which of the ultrasound parameters were independent predictors for endometriomas, a multiple regression analysis was performed (using the “enter” input model)”. What is the ““enter” input model”? In addition, regression analysis is an important part of this study and should be clearly introduced.
- In Table 3, the 95% confidence interval of specificity for “maximum of 5 locules” ranges from 86.8% to 100%, but the final specificity is 100%. How was the value “100%” calculated? Mean? Median? … Typically, the value represents the “mean” number, but this should not be 100% based on the 95% confidence interval. I think the authors need to interpret this more clearly.
- [Line 272] “The ROC analysis showed that the prediction model exceeded the diagnostic ability of all independent features in terms of both sensitivity and specificity (Table 6, Figure 2).” The authors need to test the significance of the conclusion.
Minor comments:
- [Line 227] “… age range: 22–54 years). According to their final diagnosis …” might be “… age range: 22–54 years), according to their final diagnosis”.
- [Line 242 and other places] “confidential interval” might be “confidence interval”.
Reviewer 2 Report
This is an informative article identifying key texture features that can promote a more accurate diagnosis of endometriomas and hemorrhagic ovarian cysts. My suggestions for the manuscript are as follows:
- I don’t see a table that summarizes the diagnostic model the authors developed based on texture features. Including such a table with specific criteria and diagnostic guidelines would be helpful to the medical community.
- I couldn’t find detailed descriptions of how the authors reached the conclusion of 100% sensitivity and specificity of the texture model. Because sensitivity and specificity are essential indicators of diagnostic accuracy, I recommend the authors describe at length how the two rates were derived, including how true positives were identified.
- Throughout the article, I noticed grammatical errors and awkward sentence structures. The manuscript needs to be edited by a professional editor who specializes in life sciences.
Round 2
Reviewer 1 Report
Dear editor:
The authors did a good job addressing most of the previous comments and critiques. Although, I still think my first comment might have potential for improvement based on the VIFs, the authors’ reply seems to be intuitive. In general, I think the authors have provided an excellent workflow to test the difference between the texture features of endometriomas and hemorrhagic ovarian cysts.